# An improved digestion and analysis procedure for silicon in plant tissue

**Noah James Langenfeld**\*, **Bruce Bugbee**

Crop Physiology Laboratory, Department of Plants, Soils, and Climate, Utah State University, Logan, Utah, United States of America

\* noah.langenfeld@usu.edu

## Abstract

Silicon (Si) in plant tissues reduces abiotic and biotic stress, but it is incorporated as silica ($SiO_2$), which is difficult to solubilize for analysis. We modified an oven-induced tissue-digestion and analysis method to improve Si solubilization and validated its accuracy by quantifying the mass-balance recovery of Si from the hydroponic solution and plant tissues of cucumber (*Cucumis sativus*). Leaf, stem, and root tissues were dried, finely-ground, and digested in 12.5 molar sodium hydroxide at 95˚C for 4 hours. Solutions were then acidified with 6 molar hydrochloric acid to achieve a pH below 2 for measurement of Si using the molybdate blue colorimetric method. Interference of phosphorus in the analysis was minimized by increasing the addition of oxalic acid from 0.6 to 1.1 molar. We recovered 101% ± 13% of the expected Si, calculated using mass-balance recovery, in leaf, stem, and root tissues across 15 digestions. This Si recovery was fourteen-fold higher than the standard acid-extraction method and similar to a USDA-ARS alkaline-extraction method. Our procedure offers a low-cost, accurate method for extraction and analysis of Si in plant tissues.

## Introduction

Silicon (Si) is the second largest component of Earth's crust, where it occurs as silica ($SiO_2$) instead of its free ionic form. While Si is not considered an essential element for plant growth [1], it can have many beneficial effects on plant health [2]. Silicon can increase disease resistance by physically strengthening cell walls and increasing the production of flavonoid and antimicrobial compounds [3, 4]. Some plants, such as cucumber (*Cucumis sativus*) and sunflower (*Helianthus annus*), can accumulate at least 1% Si in their leaf tissue [5], while rice (*Oryza sativa*) and sugarcane (*Saccharum officinarum*) can contain up to 10% of their dry matter as Si [6, 7].

Plants take up Si as monosilicic acid ($Si(OH)_4$) [2], and store it in the same way, as silica ($SiO_2$) in leaf cuticles, cellular lumens, and cell walls [8, 9].

Silicon must be solubilized from the plant tissue for analysis. Silica is weakly soluble up to a pH of 9, after which the solubility increases exponentially [10]. This requires digesting tissue with a strong oxidant and heat. Some methods utilize an autoclave [11] or microwave digestion system [12], but there is significant variability among methods [7].

Kraska and Breitenbeck [7] compared an oven-induced digestion (OID) method for Si extraction to autoclave-induced, modified autoclave, alkali fusion, and microwave tissue

**Data Availability Statement:** All relevant data are within the paper and its Supporting Information files.

**Funding:** This research was supported by the Utah Agricultural Experiment Station (B.B.), Utah State

University, and approved as journal paper number 9684; National Aeronautics and Space Administration (NASA, B.B.), Center for the Utilization of Biological Engineering in Space (grant number NNX17AJ31G). The funders did not and will not have a role in study design, data collection and analysis, decision to publish, or preparation of the manuscript.

**Competing interests:** The authors have declared that no competing interests exist.

digestion methods. The OID method recovered similar or slightly more Si from rice straw and sugarcane leaves than previous methods and provided less variable measurements. They also found no significant difference in Si concentrations between solutions analyzed with molybdate blue colorimetry (MBC) or inductively-coupled plasma optical emission spectroscopy (ICP-OES) if ammonium fluoride was added to improve color stability prior to quantification with MBC.

Although ICP-OES accurately quantifies elemental concentrations in the presence of interferences and in complex matrices, a high capital cost limits its use to large analytical laboratories. The MBC method can be conducted using inexpensive reagents and a colorimeter, but it is subject to interference from other elements, such as iron (Fe) and phosphorus (P). These elements react with molybdate to form complexes with similar absorbance wavebands as silcomolybdate acid [13]. The concentration of Fe in plant tissue is typically 100-fold less than Si and it thus minimally interferes with the analysis. Phosphorus is present at similar levels to Si in plant tissue [14] and can thus cause a substantial interference. Oxalic acid is typically added in the standard MBC procedure to destroy molybdate-P complexes and minimize the interference [15]. Chalmers and Sinclair [16] saw an incomplete destruction of these complexes and found tartaric acid more efficient at eliminating the P interference than oxalic acid. However, Combatt Caballero et al. [17] more recently analyzed P interference up to 1 mg $L^{-1}$ in the MBC method and found oxalic acid to be better at suppressing the interference than tartaric, citric, or malic acid.

The polymerization of silicic acid presents unique analytical challenges when using the MBC procedure. Silica reacts with water to form monosilicic acid, which can then polymerize to form polysilicic acid; however, only monosilicic acid reacts with molybdate during the MBC procedure. Monosilicic acid does not polymerize if the pH is less than 4 [18], but monosilicic acid and molybdate must be below pH 2 to facilitate complexation and color development [19]. Polymerization of monosilicic acid can increase under sodium chloride concentrations above 50 mM, but this only occurs at a pH greater than 6 [20].

Octanol is typically added to samples prior to digestion to reduce foaming, which is undesirable as it interferes with digestion completeness. Octanol is a surfactant with one of the highest foam breaking abilities among common alcohols [21]. Only a few drops are typically needed per sample vial to control foaming caused by tissue oxidation.

Ammonium fluoride stabilizes the color of the molybdo-silicate complex. Although fluoride ions can catalyze the polymerization of monosilicic acid below a pH of 2 [22, 23], Kraska and Breitenbeck [7] found that the addition of a millimolar concentration of fluoride was necessary to stabilize color development and aided in the measurement of monosilicic acid in solution.

While the OID method has been shown to recover more Si than previous methods, no analyses have been published measuring the absolute Si recovery from plant tissue. Our objective was to validate the accuracy of the oven-induced digestion procedure for extraction and colorimetric analysis of Si by using plant tissues with known Si concentrations.

## Materials and methods

We grew cucumber (*Cucumis sativus* cv. Fanfare) in deep-flow hydroponics because it is a Si accumulator [24]. The use of a mass balance approach to estimate elemental uptake allowed us to calculate the theoretical concentration of Si within each plant. The fate of silicon added to the nutrient solution was either in the nutrient solution or in the plant at harvest. This approach has been used to quantify mass-balance recovery of other elements in plant tissues from hydroponic culture [25].

The full digestion protocol described in this peer-reviewed article is published on protocols.io, https://doi.org/10.17504/protocols.io.ewov1o3e7lr2/v1, and is included for printing as

S1 File with this article. All solutions from the OID method were analyzed using a colorimeter (Smart3 Colorimeter, LaMotte, Chestertown, MD, USA).

Plant tissue was also digested by the Utah State University Analytical Laboratory in Logan, UT using method B-4.25 in [26] (standard method), and by the USDA-ARS Application Technology Research Unit worksite in Toledo, OH using the digestion method described in [25, 27] with ramp and holding times increased from 15 to 20 minutes. Tissue digestions from both methods were subsequently analyzed using ICP-OES at their respective laboratories.

## Expected results

Cucumber was grown in a deep-flow hydroponic system (Fig 1) to facilitate mass balance recovery of Si in the nutrient solution and the plant tissue [25]. The Si content in leaf, stem, and root tissues, as well as the nutrient solution, was measured at harvest using MBC. We retained the use of 5 mM ammonium fluoride, as described in [7], following tissue digestion and found no reduction in Si recovery. Silicon was added along with other nutrients in a dilute solution to the hydroponic root-zones as needed to maintain a constant solution depth. Silicon uptake was estimated to be the difference between the total Si added to the nutrient solution over the study duration and the Si remaining in solution at the end. The standard plant tissue from the National Institute of Standards and Technology of the United States (Standard Reference Material 1547 Peach Leaves) does not include a concentration for Si [28], so recovery was calculated using mass-balance principles. Comparing the sum of plant uptake with expected uptake allowed us to calculate how closely we attained the mass-balance recovery of Si (Table 1). Fifteen tissue digestions from six plants resulted in a Si mass-balance recovery of 101% ± 13%.

There is a significant economy of scale in this procedure. A single sample took 5 hours to analyze, while nine samples took 5.25 hours.

### Minimizing interference from phosphorus

The presence of P can cause an overestimation of the Si concentration. We confirmed that the P interference was not eliminated by the addition of the standard concentration of oxalic acid (0.6 M) as shown in Table 2. To further minimize P interference, we increased the concentration of oxalic acid from 0.6 to 1.1 M. Interference of P was reduced to 0.06 equivalents of silica (less than 4%). Further increases in oxalic acid concentration may be difficult because the maximum solubility is about 1.3 M at 25˚C.

### Determining silicon tissue content

We first subtracted the colorimetric value for a deionized water blank from a sample measurement. This corrected value was multiplied by the volume of our digestion container after its final dilution (0.05 L). We divided this value by the sample mass (about 100 mg or 0.0001 kg) to calculate the silica concentration in our sample vial. This value was then multiplied by 0.467 (the ratio of the molar mass of Si to silica) to convert silica into Si, and further multiplied by 25 to account for sample dilution immediately prior to measurement with MBC. An example of this calculation is shown in Eq 1.

$$\frac{\frac{(1.09-0.25)\ mg\ SiO_2}{L} \times (0.05)\ L}{(0.0001)\ kg} \times 0.467 \times 25 = 4,908 \frac{mg}{kg} Si = 0.49\%\ Si \tag{1}$$

The average Si content for cucumber across the entire plant was 0.49 ± 0.1% among the 12 replicate digestions from Table 1, which is typical of a Si-accumulating species.

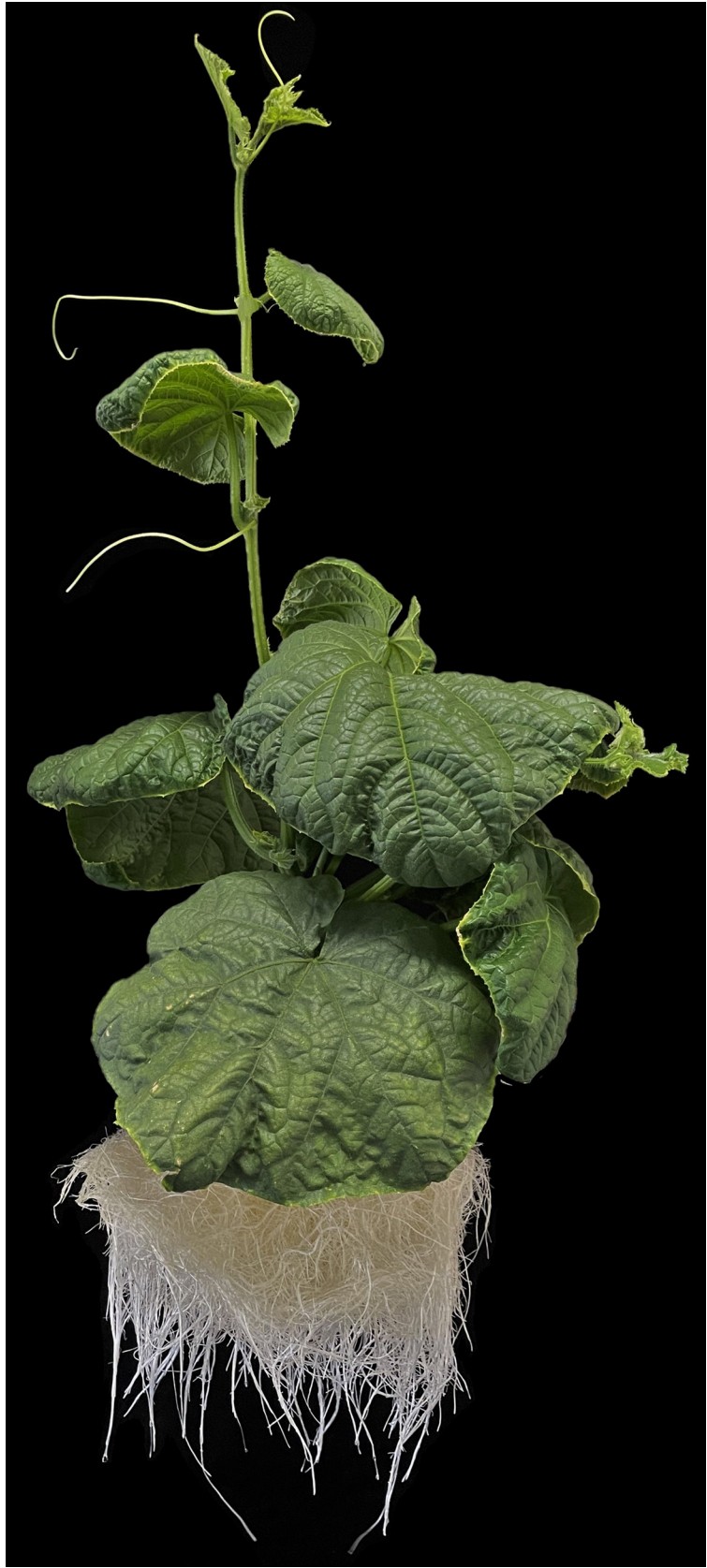

**Fig 1. An example cucumber (*Cucumis sativus* cv. Fanfare) plant grown in deep-flow hydroponics prior to analysis of tissue content for Si.**

**Table 1. Silicon recovery from four replicate digestions of leaf, stem, and root tissues, and the nutrient solution at harvest of three cucumber (*Cucumis sativus* cv. Fanfare) plants grown in deep-flow hydroponics.** Values have been rounded for readability; actual recoveries are shown on the last line of the table.

| | Replicate plant | | |
|---|---|---|---|
| **Tissue content (mg Si per plant)** | **1** | **2** | **3** |
| Leaf | 108 | 108 | 168 |
| Stem | 6 | 11 | 13 |
| Root | 11 | 11 | 16 |
| **Total** | **125** | **130** | **197** |
| Nutrient solution (mg Si) | | | |
| Total Si added to solution | 204 | 189 | 209 |
| Si remaining at harvest | 82 | 48 | 29 |
| **Amount removed from solution** | **122** | **141** | **180** |
| **Percent recovery in plant tissues (%)** | **101** | **94** | **108** |

**Table 2. The effect of oxalic acid on phosphate interference in the colorimetric analysis of Si.** Without oxalic acid, phosphorus (P) interference caused the reading to over-range (4 ppm silica equivalents). The addition of 0.6 M oxalic acid reduced the interference to 36%, and at 1.1 M oxalic acid interference was 4% (0.06 / 1.49). This test included a high P background. Lower P background levels would have less interference.

| Oxalic acid | $1.49$ mg L$^{-1}$ silica with $1.8$ mg L$^{-1}$ phosphate background | Interference from phosphate |
|---|---|---|
| (M) | (mg L$^{-1}$ silica) | (mg L$^{-1}$ silica) |
| 0 | More than 4 (over-range) | — |
| 0.60 | 2.03 | 0.54 (36%) |
| 1.1 | 1.43 | 0.06 (4%) |

## The importance of sample grinding

Complete sample grinding is critical to recovery of Si in leaf tissue. Large particle sizes are more difficult to digest. The mean coefficient of variation for Si content by tissue was 51% for stems, 19% for leaves, and 13% for roots (Table 3) Stems are difficult to grind and these results suggest that further grinding of stems may reduce the variability in Si quantification.

Ground tissue samples can be stored indefinitely at room temperature if kept dry, but digested samples should be analyzed the same day as their digestion. We found increased variability in Si concentrations if samples digested using OID were analyzed via MBC more than a day after being digested, even if they were stored under refrigeration at 4°C. We do not have evidence for a mechanism of change during storage and caution against storing samples for long periods of time prior to analysis. Acidifying samples with a strong acid may increase storage lifetime.

## Comparison with two tissue digestion methods

Common tissue digestion methods do not always completely solubilize and extract Si from plant tissue (Table 4). The nitric acid-extraction method [26] used by many laboratories for

**Table 3. Coefficient of variation for the concentration of Si among four replicate digestions of leaf, stem, and root tissues in three cucumber (*Cucumis sativus* cv. Fanfare) plants.**

| | Coefficient of variation per plant (%) | | | Average (%) |
|---|---|---|---|---|
| **Tissue** | **1** | **2** | **3** | |
| Leaf | 9 | 32 | 17 | 19 |
| Stem | 40 | 47 | 65 | 51 |
| Root | 15 | 14 | 11 | 13 |

**Table 4. A comparison of Si concentrations of three cucumber (*Cucumis sativus* cv. Fanfare) plants digested using an improved OID extraction, acid-extraction, or USDA alkaline-extraction.** Improved OID extraction values are from independent sets of plant tissue.

| | Si in plant tissue (mg Si kg$^{-1}$) | | | | | | | | |
|---|---|---|---|---|---|---|---|---|---|
| Tissue | Leaf | | | Stem | | | Root | | |
| Plant | 1 | 2 | 3 | 1 | 2 | 3 | 1 | 2 | 3 |
| Standard acid extraction | 664 | 843 | 853 | 586 | 737 | 709 | 476 | 432 | 390 |
| Improved OID extraction | 5685 | 7115 | 7659 | 1381 | 2077 | 1876 | 1874 | 2127 | 1974 |
| Standard as % of improved OID | 12 | 12 | 11 | 42 | 35 | 38 | 25 | 20 | 20 |
| USDA alkaline extraction | 4720 | 6207 | 5760 | 911 | 914 | 921 | 990 | 1043 | 1273 |
| Improved OID extraction | 6917 | 5648 | 5915 | 3247 | 2402 | 3914 | 2357 | 1557 | 2846 |
| Alkaline as % of improved OID | 68 | 110 | 97 | 28 | 38 | 24 | 42 | 67 | 45 |

**Table 5. Mean and standard deviation of the mass balance recovery of Si measured with the standard acid method, the improved OID method, and the USDA alkaline-extraction method.** The improved OID method had a 14-fold higher recovery of Si than the standard acid method, and a 7% higher recovery than the USDA alkaline method. The standard deviation was less than 7% for all methods.

| | Total Si recovery (%) | |
|---|---|---|
| Method | Mean | Standard deviation |
| Standard acid | 7 | 0.6 |
| Improved OID | 101 | 7 |
| USDA alkaline | 92 | 3 |
| Improved OID | 99 | 6 |

quantification of plant macro- and micronutrients resulted in a mass balance recovery of only 7% ± 0.6%. The USDA alkaline-extraction method [27] had a mass balance recovery of 92% ± 3%, which was statistically similar to the mass balance recovery of 99% ± 6% achieved using the OID method (Table 5). The USDA method recovered similar amounts of Si in leaf tissue but may have underestimated Si in stem and root tissue. The Si may be in more recalcitrant forms in stems and roots, and thus more difficult to solubilize in these tissues.

This improved OID method of Si extraction and analysis in plant tissue does not require expensive reagents or analytical instrumentation. The high total percent recovery of Si in cucumber tissue demonstrates the accuracy of the method. This will be useful to growers and researchers looking to analyze Si in plant tissue.

## Supporting information

**S1 File. Step-by-step protocol, also available on protocols.io.**
(PDF)

## Acknowledgments

We acknowledge Hikari Ai Skabelund and Mackenzie Dey for their assistance in the early development and refinement of this protocol. We are also grateful to Jennifer Boldt at the USDA-ARS Application Technology Research Unit and Tiffany Evans at the Utah State University Analytical Laboratory for assistance in digesting and analyzing tissue samples using their labs' respective methods. We additionally thank Jennifer Boldt for her kind internal review of this manuscript.

## Author Contributions

**Data curation:** Noah James Langenfeld.

**Formal analysis:** Noah James Langenfeld.

**Funding acquisition:** Bruce Bugbee.

**Investigation:** Noah James Langenfeld.

**Methodology:** Noah James Langenfeld.

**Project administration:** Bruce Bugbee.

**Supervision:** Bruce Bugbee.

**Writing – original draft:** Noah James Langenfeld.

**Writing – review & editing:** Noah James Langenfeld, Bruce Bugbee.

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
