## [Decision Letter · Decision Letter 0]

8 May 2023

PONE-D-23-10481An improved digestion and analysis procedure for silicon in plant tissuePLOS ONE

Dear Dr. Langenfeld,

Thank you for submitting your manuscript to PLOS ONE. After careful consideration, we feel that it has merit but does not fully meet PLOS ONE’s publication criteria as it currently stands. Therefore, we invite you to submit a revised version of the manuscript that addresses the points raised during the review process.

We look forward to receiving your revised manuscript.

Kind regards,

Rupesh Kailasrao Deshmukh, Ph.D.

Academic Editor

PLOS ONE

Journal Requirements:

2. Please amend either the title on the online submission form (via Edit Submission) or the title in the manuscript so that they are identical.

3. We note you have not yet provided a protocols.io PDF version of your protocol and/or a protocols.io DOI. When you submit your revision, please provide a PDF version of your protocol as generated by protocols.io (the file will have the protocols.io logo in the upper right corner of the first page) as a Supporting Information file. The filename should be S1_file.pdf, and you should enter “S1 File” into the Description field. Any additional protocols should be numbered S2, S3, and so on. Please also follow the instructions for Supporting Information captions [https://journals.plos.org/plosone/s/supporting-information#loc-captions]. The title in the caption should read: “Step-by-step protocol, also available on protocols.io.”

Please assign your protocol a protocols.io DOI, if you have not already done so, and include the following line in the Materials and Methods section of your manuscript: “The protocol described in this peer-reviewed article is published on protocols.io (https://dx.doi.org/10.17504/protocols.io.[...]) and is included for printing purposes as S1 File.” You should also supply the DOI in the Protocols.io DOI field of the submission form when you submit your revision.

If you have not yet uploaded your protocol to protocols.io, you are invited to use the platform’s protocol entry service [https://www.protocols.io/we-enter-protocols] for doing so, at no charge. Through this service, the team at protocols.io will enter your protocol for you and format it in a way that takes advantage of the platform’s features. When submitting your protocol to the protocol entry service please include the customer code PLOS2022 in the Note field and indicate that your protocol is associated with a PLOS ONE Lab Protocol Submission. You should also include the title and manuscript number of your PLOS ONE submission.

Additional Editor Comments:

The reviewers have identified several areas that require extensive revision in the article, and it is crucial for the authors to address these issues. In addition, the authors should restructure certain paragraphs and provide additional details to enhance the clarity of the method.

Reviewers' comments:

Reviewer's Responses to Questions

**Comments to the Author**

1. Does the manuscript report a protocol which is of utility to the research community and adds value to the published literature?

Reviewer #1: Yes

Reviewer #2: Yes

2. Has the protocol been described in sufficient detail?

To answer this question, please click the link to protocols.io in the Materials and Methods section of the manuscript (if a link has been provided) or consult the step-by-step protocol in the Supporting Information files.

The step-by-step protocol should contain sufficient detail for another researcher to be able to reproduce all experiments and analyses.

Reviewer #1: No

Reviewer #2: Partly

3. Does the protocol describe a validated method?

Reviewer #1: No

Reviewer #2: Yes

4. If the manuscript contains new data, have the authors made this data fully available?

Reviewer #1: Yes

Reviewer #2: Yes

**5. Is the article presented in an intelligible fashion and written in standard English?**

Reviewer #1: Yes

Reviewer #2: Yes

6. Review Comments to the Author

Reviewer #1: This is very interesting method paper; based on the offered analytical results, it seems a valid method. Yet it only focuses on one plant, cucumber. Thus, I am questioning its wide use to other plants. Does it keep the same valid result and recovery rate when it is used to other? This means that the current manuscript should carefully and clearly fix the cucumber, not confusing its use to other plants. In this case, much better to change its tittle with the analysis of cucumber plant. An improved digestion and analysis procedure for silicon in cucumber plant tissue No？

In addition, does this method save lots of time when analyzing lots of plant samples? Or it is much better to consider small amounts of plant samples?

Reviewer #2: The article entitled "An improved digestion and analysis procedure for silicon in plant tissue" is well written. This article will help researchers in the low-cost estimation of Si by applying this information to their respective research. The MS has analyzed the different Si estimation approaches by comparing them. The article is well organized with figures and tables. However, I have some corrections and suggestions for the author.

Line 33- Can you please make the statement more clear that “Is Si the largest component on earth’s crust” or “abundantly available component on earth’s crust?

Line 38- You can either use silica or SiO2. If you're using SiO2, follow the same trend throughout your MS.

Line 39-40- Provide reference

Line 57-59- Rewrite the paragraph and try to merge it with the next one.

Line 64 and 66- Provide reference

Line 71- 73- Rewrite the whole sentence.

Line 77-79- Merge the paragraph with the previous one.

Line 90-91- Consider revision of this sentence for clarity

Line 109-110- How many times do you change the nutrient solution of the hydroponic system, similarly how many times the silicon was provided in the nutrient solution? How did you calculate the total silicon content of the medium? It would be good if authors can write a few more lines about it.

Line 120-121- It would be good if the author can explain how they confirmed the elimination of P interference.

Line 135- Please write a few lines about why you multiplied it with 0.467.

Line 150-152- It would be good if the author can highlight the probable reasons behind this.

7. PLOS authors have the option to publish the peer review history of their article (what does this mean?). If published, this will include your full peer review and any attached files.

Reviewer #1: No

Reviewer #2: No

---

## [Author Response · Author response to Decision Letter 0]

26 May 2023

Dear PLOS ONE Editor and Reviewers,

We sincerely appreciate the time you took to review our article. Peer reviews are essential to improving our science. Please see below for a point-by-point response to the questions raised during the review process.

Editor

1. Our manuscript meets PLOS ONE’s style requirements to the best of our knowledge.

2. The title on the online submission form and the title in the manuscript do match. The title is “An improved digestion and analysis procedure for silicon in plant tissue”. Line 2 of the manuscript is a short title, as indicated by the PLOS ONE website (https://journals.plos.org/plosone/s/submission-guidelines#:~:text=Include%20a%20full%20title%20and%20a%20short%20title%20for%20the%20manuscript)

3. We already completed and uploaded a PDF copy of the protocols.io version. This was included as S1_file.pdf in the original submission. The DOI was generated and was already included in the submission manuscript in both the Associated Content and Materials and Methods Section.

Reviewer 1

1. We focused on cucumber in this manuscript because it is a Si-accumulating species that is also commonly grown in controlled environments. The forms of silica in plant tissue are not species dependent, so we have no reason to believe that these results would differ among other species. The original paper this work is based on, for example, tested both rice and sugarcane, and found a high Si recovery in both species.

2. This method can be conducted with one sample or dozens of samples. As with most analytical tests, time efficiency increases with increasing sample size. This method is therefore no different in its time efficiency than comparative methods. While there are longer heating times, researchers do not have to be present during these times for monitoring, which could potentially increase time efficiency.

Reviewer 2

Line 33: We modified the first sentence of the introduction to say, “Si is the second largest component in the Earth’s crust”. We removed any reference to oxygen, and thus removed confusion.

Line 38: Excellent comment. We have revised the manuscript to maintain consistency of referencing silica vs SiO2. Silica is chemically specified as SiO2 upon first mention in the manuscript, and it is referred to as silica in subsequent mentions.

Line 39-40: Thank you for this comment. We have added a reference concerning the forms and locations of Si in plants.

Line 57-59: Thank you for this suggestion. We have rewritten this paragraph and have merged it with the subsequent paragraph.

Line 64: We have added a reference to this sentence that provides average concentrations for phosphorus in plant tissue.

Line 66: We have added a reference on the standard colorimetric methods.

Line 71-73: We have rewritten the sentence to aid in clarity.

Line 77-79: This paragraph represents a separate idea from the previous paragraph. So, while short, we felt it has merit to stand on its own.

Line 90-91: We have edited this sentence for clarity.

Line 109-110: Thank you for your question. We have rewritten this paragraph to clarify how the Si was added over the study. Following the principles of mass balance cited earlier in the manuscript, we refill with a dilute nutrient solution throughout the study as needed and accurately quantify how much Si was added to the root-zone over the study.

Line 120-121: This data is shown in table 2. We have rewritten the sentence to specify this.

Line 135: We agree this needed to be clarified. We added a phrase to the sentence to specify how the number 0.467 was derived.

Line 150-152: We do not have a solid reason why solutions become less stable over time. We thus rewrote this section to specify this issue.

Thank you!

Noah J. Langenfeld and Bruce Bugbee

Crop Physiology Laboratory – Utah State University

---

## [Decision Letter · Decision Letter 1]

7 Jul 2023

PONE-D-23-10481R1An improved digestion and analysis procedure for silicon in plant tissuePLOS ONE

Dear Dr. Langenfeld,

Thank you for submitting your manuscript to PLOS ONE. After careful consideration, we feel that it has merit but does not fully meet PLOS ONE’s publication criteria as it currently stands. Therefore, we invite you to submit a revised version of the manuscript that addresses the points raised during the review process. Please submit your revised manuscript by Aug 21 2023 11:59PM. If you will need more time than this to complete your revisions, please reply to this message or contact the journal office at plosone@plos.org. Please include the following items when submitting your revised manuscript:A rebuttal letter that responds to each point raised by the academic editor and reviewer(s). You should upload this letter as a separate file labeled 'Response to Reviewers'.A marked-up copy of your manuscript that highlights changes made to the original version. You should upload this as a separate file labeled 'Revised Manuscript with Track Changes'.An unmarked version of your revised paper without tracked changes. You should upload this as a separate file labeled 'Manuscript'.If applicable, we recommend that you deposit your laboratory protocols in protocols.io to enhance the reproducibility of your results. Protocols.io assigns your protocol its own identifier (DOI) so that it can be cited independently in the future. For instructions see: https://journals.plos.org/plosone/s/submission-guidelines#loc-laboratory-protocols. Additionally, PLOS ONE offers an option for publishing peer-reviewed Lab Protocol articles, which describe protocols hosted on protocols.io. Read more information on sharing protocols at https://plos.org/protocols?utm_medium=editorial-email&utm_source=authorletters&utm_campaign=protocols.

We look forward to receiving your revised manuscript.

Kind regards,

Rupesh Kailasrao Deshmukh, Ph.D.

Academic Editor

PLOS ONE

Journal Requirements:

Additional Editor Comments:

One of the reviewers expressed dissatisfaction with the authors' response to a specific comment. In order to address this concern, I highly recommend that the authors provide a more detailed clarification in their response. Additionally, it would be beneficial for them to further polish their manuscript to enhance its overall quality and ensure that any potential issues or ambiguities are thoroughly addressed.

Reviewers' comments:

Reviewer's Responses to Questions

**Comments to the Author**

1. Does the manuscript report a protocol which is of utility to the research community and adds value to the published literature?

Reviewer #1: Yes

Reviewer #2: Yes

2. Has the protocol been described in sufficient detail?

To answer this question, please click the link to protocols.io in the Materials and Methods section of the manuscript (if a link has been provided) or consult the step-by-step protocol in the Supporting Information files.

The step-by-step protocol should contain sufficient detail for another researcher to be able to reproduce all experiments and analyses.

Reviewer #1: No

Reviewer #2: Yes

3. Does the protocol describe a validated method?

Reviewer #1: No

Reviewer #2: Yes

4. If the manuscript contains new data, have the authors made this data fully available?

Reviewer #1: No

Reviewer #2: N/A

**5. Is the article presented in an intelligible fashion and written in standard English?**

Reviewer #1: Yes

Reviewer #2: Yes

6. Review Comments to the Author

Reviewer #1: Authors’ response: 1.We focused on cucumber in this manuscript because it is a Si-accumulating species

that is also commonly grown in controlled environments. The forms of silica in plant

tissue are not species dependent, so we have no reason to believe that these results

would differ among other species. The original paper this work is based on, for

example, tested both rice and sugarcane, and found a high Si recovery in both

species.

Reviewer #1. Here, I think that author cannot clearly get the previous comments.

I clearly understand this protocol focused on the cucumber which is also a Si-rich accumulator. This is why I suggest that author rephase the title of this manuscript as ‘ An improved digestion and analysis procedure for silicon in cucumber tissue: Digestion of silicon in its tissue’. This can help reader and future researchers clearly and precisely to use this protocol for other plants and their tissues.

If author carried out different plant species, the analytical results illustrate the same Si recovery, which can suggest that it can be widely used to other crops or plants. A scientific literature should be precise and reasonable, especially for a standard method/protocol for our scientific issue.

Authors’ response: 2.This method can be conducted with one sample or dozens of samples. As with most

analytical tests, time efficiency increases with increasing sample size. This method is therefore no different in its time efficiency than comparative methods. While there are longer heating times, researchers do not have to be present during these times for

monitoring, which could potentially increase time efficiency.

Reviewer #1. Did you have any data to support their time efficiency? Is it the same efficiency for the precise silicon content? Fast is not best but should be precise. If not, how can we use this protocol to compare to previous findings?

Reviewer #2: No comments, Authors addressed all my comments satisfactorily, I recommend this article for publication

7. PLOS authors have the option to publish the peer review history of their article (what does this mean?). If published, this will include your full peer review and any attached files.

Reviewer #1: No

Reviewer #2: **Yes: **Gaurav Raturi

---

## [Author Response · Author response to Decision Letter 1]

11 Jul 2023

Reviewer 1

1. Thank you for your comments about including the species name in the title. We did an additional literature review to determine the form of Si that is stored in plant tissue. Two manuscripts have found that Si is always stored as silica regardless of species. We reorganized paragraphs in the manuscript and cited these references:

“Plants take up Si as monosilicic acid (Si(OH)4) [2], and store it in the same way, as silica (SiO2) in leaf cuticles, cellular lumens, and cell walls [7,8].”

[2] Luyckx M, Hausman J-F, Lutts S, Guerriero G. Silicon and plants: Current knowledge and technological perspectives. Front Plant Sci. 2017;8. doi:10.3389/fpls.2017.00411

[7] Sangster AG, Hodson MJ, Tubb HJ. Silicon deposition in higher plants. Studies in Plant Science. Elsevier; 2001. pp. 85–113. doi:10.1016/S0928-3420(01)80009-4

[8] Lanning FC, Ponnaiya BWX, Crumpton CF. The Chemical Nature of Silica in Plants. Plant Physiol. 1958;33: 339–343. doi:10.1104/pp.33.5.339

Since the storage form is uniform among species, extraction should be the same among species. We would like to keep the word ‘cucumber’ out of the title as we feel including it would limit the article impact. The abstract does clearly indicate that the work was done on cucumber because it has high concentration of Si in its tissues. 

2. Thank you for your concern about the test length efficiency. We added a paragraph indicating the time it took to run a single sample and multiple samples:

“There is a significant economy of scale in this procedure. A single sample took 5 hours to analyze, while nine samples took 5.25 hours.”

Reviewer 2

1. Thank you for recommending our manuscript for publication!

---

## [Editor Report · Decision Letter 2]

12 Jul 2023

An improved digestion and analysis procedure for silicon in plant tissue

PONE-D-23-10481R2

Dear Dr. Langenfeld,

We’re pleased to inform you that your manuscript has been judged scientifically suitable for publication and will be formally accepted for publication once it meets all outstanding technical requirements.

Kind regards,

Rupesh Kailasrao Deshmukh, Ph.D.

Academic Editor

PLOS ONE
---

## [Editor Report · Acceptance letter]

14 Jul 2023

PONE-D-23-10481R2 

An improved digestion and analysis procedure for silicon in plant tissue 

Dear Dr. Langenfeld:

I'm pleased to inform you that your manuscript has been deemed suitable for publication in PLOS ONE. Congratulations! Your manuscript is now with our production department. 

Kind regards, 

on behalf of

Dr. Rupesh Kailasrao Deshmukh 

Academic Editor

PLOS ONE